# An Ontology Enrichment Framework Using Retrieval-Augmented Large Language Models

## Abstract

Ontology enrichment, understood as the process of extending and refining existing ontologies with new concepts, relations, and instances, has become a critical task for building robust and up-to-date knowledge bases. The exponential growth of scientific publications, datasets, and multimodal resources makes manual enrichment highly impractical, creating the need for automated or semi-automated approaches. In this work, we propose a framework that leverages multimodal large language models and retrieval-augmented generation to support ontology enrichment. Our method systematically extracts semantic knowledge units, aligns them with existing ontological structures, and generates interlinked triples, thereby enhancing both the coverage and the expressivity of the ontology. This framework addresses the knowledge acquisition bottleneck by enabling scalable integration of heterogeneous resources and fostering cross-domain semantic interoperability. To illustrate its effectiveness, we apply the framework to the domain of 4D printing, a rapidly evolving field at the intersection of materials science, manufacturing, and design. By incorporating knowledge about materials, properties, stimuli interactions, process parameters, and design strategies, the framework enriches a domain-specific ontology and supports innovation in the development of programmable and multifunctional structures. The proposed framework follows a four-stage pipeline that combines multimodal retrieval of relevant text and figures from scientific literature with the ingestion of structured datasets and existing knowledge graphs, uses a fine-tuned multimodal LLM to extract ontology-aligned triplets, applies multi-criteria validation based on semantic relevance and consistency, and finally performs ontology population through symbolic reasoning.

## 1 Introduction

In the era of artificial intelligence (AI) and data-driven technologies, the ability to structure and interpret knowledge has become a cornerstone of intelligent systems. While vast amounts of data are continuously generated, their utility depends on transforming raw information into machine-readable semantic representations. Ontologies have emerged as a key solution to this challenge, providing a formal and explicit specification of a shared conceptualization of a domain (Gruber, 1993). They allow the definition of concepts, properties, and semantic relations, which enables reasoning, knowledge integration, and inference beyond the explicitly available information (Guarino et al., 2009). Ontologies play a central role in the development of the Semantic Web, where they serve as the backbone for annotating and linking web resources with machine-interpretable semantics (Shadbolt et al., 2006). Instead of being limited to unstructured or human-centered information, the Semantic Web envisions a knowledge-rich environment where data can be shared, reused, and reasoned upon across heterogeneous systems. This has led to a significant research focus on domain- and task-specific ontologies, which are increasingly applied in diverse fields such as biomedicine (Bodenreider, 2004), materials science (Ghedini et al., 2017), and manufacturing (Chungoora et al., 2013). Similarly, the HERMES (spatiotemporal semantics and logical knowledge description of mecHanical objEcts in the era of 4D pRinting and programmable Matter for nExt-generation of CAD systemS) domain ontology has been established to capture 4D printing knowledge at the part design level (Dimassi et al., 2021).

Despite their structured nature, traditional ontologies are limited in dynamically adapting to evolving knowledge and in processing unstructured textual data and natural language inputs. These shortcomings highlight the need for enhanced integration between ontological systems and AI, particularly through natural language processing (NLP) and machine learning (ML) techniques (Li, 2018). Scaling LLMs has led to emergent reasoning capabilities, including in-context learning (ICL) (Peng et al., 2023), chain-of-thought (CoT) (Wei et al., 2022), and retrieval-augmented generation (RAG) (Gao et al., 2023). These advances mitigate some limitations of conventional AI models by enabling real-time knowledge retrieval and contextual inference. Additionally, the recent development of multimodal LLMs (MLLMs) has further expanded AI's ability to integrate textual, visual, and symbolic information (Yin et al., 2024). These models are particularly relevant for domains like 4D printing, where information must be captured across modalities to support design synthesis.

Through these advancements, LLMs remain fundamentally limited in interpretability and domain specificity. Their probabilistic nature can lead to hallucinations and unreliable outputs, particularly in highly specialized fields like 4D printing. Ontologies, by contrast, offer structured and interpretable knowledge representation but lack adaptability. The fusion of both technologies presents a promising approach to overcoming these challenges, especially in the enrichment of ontological data structures, termed as ontology learning. As manual annotation is labor-intensive and not scalable for large datasets or rapidly changing domains, semi-automatic methods, such as Phrase2Onto (Pour et al., 2023), have been developed by suggesting new concepts through phrase-based topic modeling; however, they still rely heavily on user input for validation, introducing potential subjectivity and inconsistency. Fully automated approaches using NLP and ML expedite the ontology extension process but are dependent on the quality of training data. These may introduce biases or errors if the data or models are not well-aligned with domain specifics. Advanced systems like online clustering with LLM agents (Wu et al., 2024) provide innovative ways to integrate new knowledge without extensive annotated datasets. However, they struggle with maintaining consistency and effectively integrating diverse information streams, posing challenges in ensuring the accuracy and relevance of ontology extensions.

The emergence of 4D printing – a technology combining smart materials and additive manufacturing (AM) – has opened new frontiers in fields requiring adaptive, deployable, or transformative structures (Demoly & André, 2022a;b). This paradigm enables objects to self-transform in response to external stimuli such as heat, light, moisture, solvent, or magnetic/electric fields (Tibbits, 2013; Ge et al., 2013). The scientific landscape of 4D printing is both rapidly evolving and inherently multidisciplinary, encompassing fields such as materials science, chemistry, mechanical engineering, process engineering, and biomimicry (Demoly et al., 2021). Since its inception in 2013, the field has experienced exponential growth, with more than 3,500 publications and an estimated annual growth rate of approximately 40%, according to the Web of Science database (Demoly & André, 2021; Demoly & André, 2021; 2024). Key challenges in advancing 4D printing include improving the printability of smart materials, enhancing their mechanical and actuation performance, promoting safe and sustainable deployment, and ensuring reliability under cyclic stimuli and real-world conditions (Demoly et al., 2021). These challenges can be considered as interdependent, especially when designing and developing practical 4D-printed systems, where trade-offs between material properties, process parameters, and functional requirements must be carefully balanced (Demoly et al., 2021). To support collective and coherent progress, it becomes vital to establish a comprehensive and dynamic knowledge and data infrastructure capable of integrate both historical findings and emerging research. Such an infrastructure is crucial for consolidating the existing body of knowledge and effectively guiding future developments.

The proposed retrieval-augmented MLLMs framework aims to integrate ontology-based reasoning with the generative and retrieval capabilities of MLLMs to support knowledge discovery across diverse domains. By embedding ontological structures within LLM architectures, the framework enhances knowledge extraction, semantic reasoning, and adaptive learning from both structured and unstructured data sources, ranging from scientific literature and datasets. This active ontology enrichment approach ensures real-time alignment with emerging research and technological advancements. To demonstrate its applicability, we apply this framework to the domain of 4D printing, where it enables the integration of cross-disciplinary insights related to smart materials, processes, and programmable structures.

## 2 ONTOLOGY ENRICHMENT FRAMEWORK

Ontology enrichment enables the enhancement of an existing preliminary ontology by automatically adding new concepts (also considered as knowledge), relationships, and individuals (meaning information or data) to make it more comprehensive and practical for a specific domain or task. To ensure both the enrichment and population of the initial ontology, we employ an integrated framework combining information retrieval with advanced text generation capabilities (as illustrated in **Figure 1**).

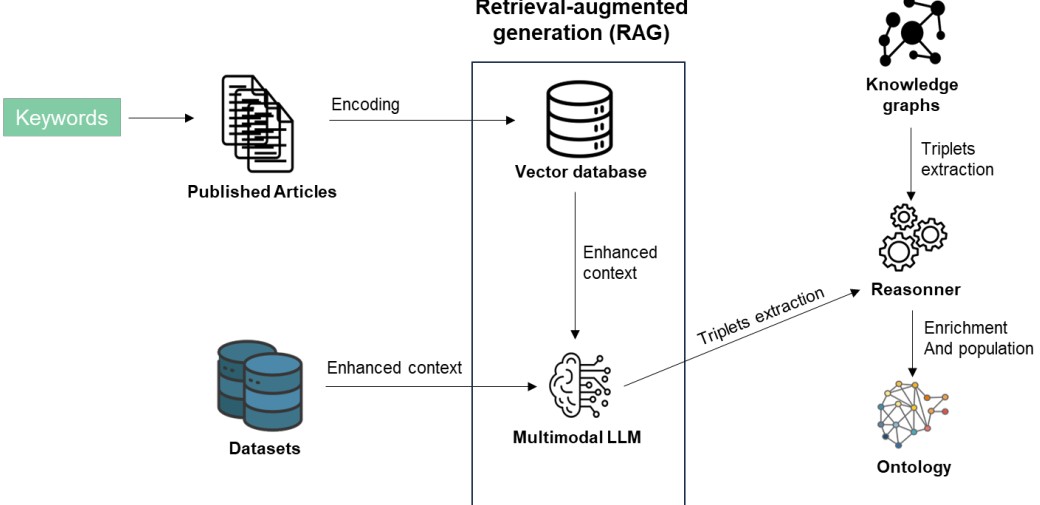

Figure 1: Retrieval-augmented ontology construction pipeline (adapted from (Bougzime et al., 2025b)).

Initially, we collected a curated corpus of published articles and domain-specific datasets using targeted keywords. Each published article is split into discrete text segments and extracted figures, while each dataset table is parsed into individual records. All text and image snippets are then encoded using a fine-tuned MLLM into dense vectors and stored in a high-throughput vector index. At inference time, the LLM issues similarity queries against this index to retrieve the top-k relevant passages or images, which it incorporates as "context windows" into its prompts. From the generated and context-aware outputs, a downstream triplet-extraction module identifies candidate [Subject-Predicate- Object] facts. These facts are merged with existing knowledge from knowledge graphs and passed to a symbolic reasoner, which enforces ontology schema constraints, checks for logical consistency, and removes duplicates. Resulted triplets are then translated into classes, properties, or instances, thereby populating and enriching the initial ontology in a continuous loop that keeps our knowledge base both up to date and semantically rigorous.

### 2.1 ONTOLOGY ENRICHMENT FROM SCIENTIFIC LITERATURE

To enrich the ontology, the process begins with the identification and selection of key terms relevant to the domain of interest. Using the ResearchRabbit application tool (res, 2025), an AI-supported scholarly discovery platform, the pertinent intersections among these keywords serve as the basis for collecting a large body of published research.

Then, we split these published articles using tools like LLM Sherpa (Nlm-atics, 2024) for robust text extraction and semantic chunking, which divided each paper into coherent chunks based on structural elements. This approach was designed to optimize both semantic completeness and computational efficiency, ensuring that each segment retained meaningful contextual information. Chunk boundaries followed the natural discourse flow (e.g., paragraphs or logical sections) rather than fixed lengths, thereby preserving local coherence throughout the segmentation process. The Aspose tool (Aspose, 2024) was used for image extraction in order to isolate each figure into standalone image files. Each token was then embedded using BERT model and CLIP for images. This process con-

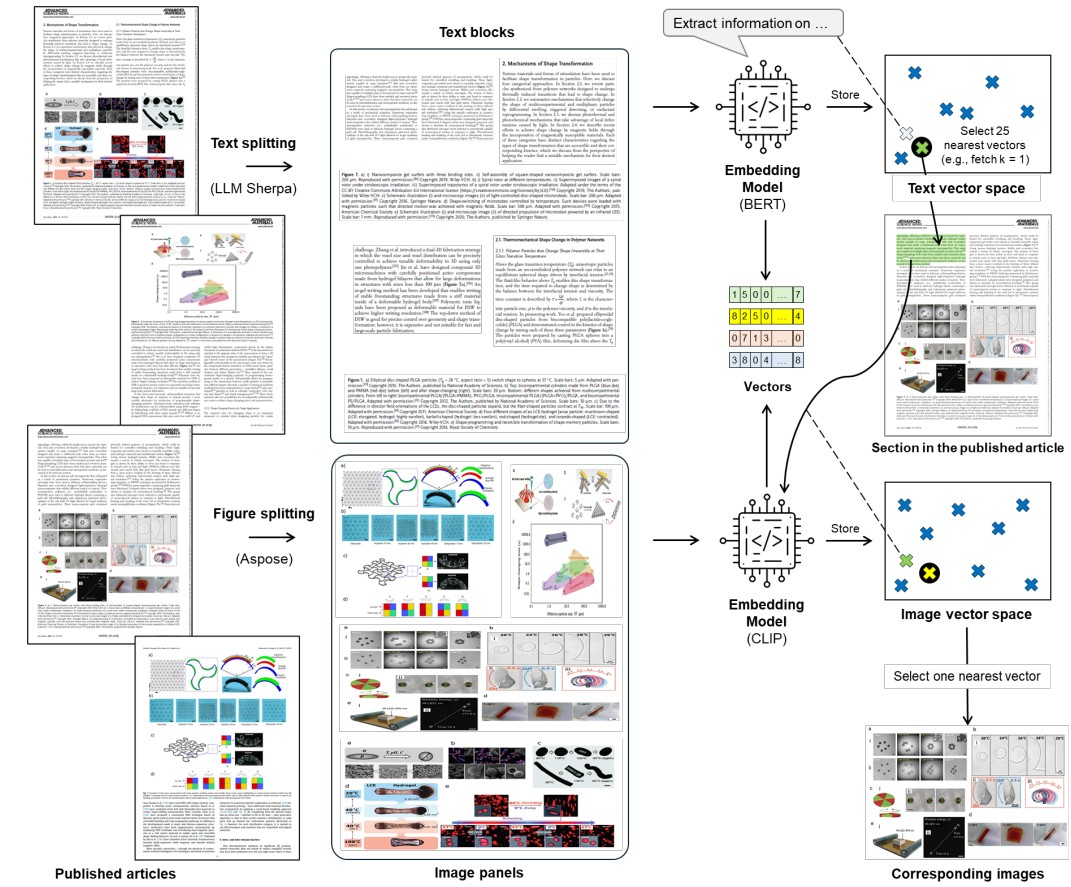

Figure 2: Pipeline for extracting textual sections and associated figures from scientific literature (adapted from (Bougzime et al., 2025b)).

verts the content into vector representations, allowing us to store them in a vector space, as shown in **Figure 2**.

Subsequently, detailed information concerning domain-specific entities, processes, and methodologies is systematically extracted from textual sources. Queries are encoded within a text vector space using BERT embeddings to identify the 25 nearest vectors. As illustrated in **Figure 2**, selecting one text vector highlights the section in green as the most relevant to the query. This section serves to identify and retrieve its corresponding relevant image through the CLIP embedding model and image vector space.

By pairing each textual section with its corresponding image, we enriched the LLaVA (Large Language-and-Vision Assistant) MLLM's input context (Parthasarathy et al., 2024; Kim et al., 2023; Jeong, 2024). This RAG process combines information retrieval with text generation, so that it helps to address challenges such as hallucination, outdated knowledge, and opaque reasoning in language models. By incorporating data from external databases, RAG ensures more accurate and credible output, particularly in knowledge-intensive tasks. This integration facilitates ongoing updates and the inclusion of specialized information, therefore making RAG a dynamic solution that combines intrinsic model knowledge with extensive external data.

Initially, we employed a few-shot learning strategy (Brown et al., 2020; Hoffmann et al., 2022; Yang et al., 2022); however, this approach proved inefficient due to the long context windows required, leading to high computational and memory costs as well as imprecise extractions. To improve efficiency and robustness, we fine-tuned the LLaVA model specifically for ontology-aligned triplet extraction (Ghanem & Cruz, 2024; Liu et al., 2022; Zhang et al., 2024), using Low-Rank Adaptation

(LoRA) (**?**) as a parameter-efficient fine-tuning (PEFT) method (Lialin et al., 2023). LoRA significantly reduces the computational footprint of fine-tuning, allowing us to specialize the model for our domain without full model retraining. This process involved embedding domain-specific knowledge within the LLaVA model, thus configuring the output format appropriately and ensuring consistent performance without the need for additional tokens. During fine-tuning, domain-specific knowledge was embedded into LLaVA so that the model could reliably distinguish ontology classes, data properties, object properties, and instances, while also preserving hierarchical constraints compliant with OWL formalism (**?**Val-Calvo et al., 2025; Doumanas et al., 2025). This approach aims to refine a multimodal large language model into a tool capable of identifying ontology-relevant triplets within a specific domain. Additional implementation details and hyperparameters are provided in Appendix C.

To enable fine-tuning, a synthetic dataset is generated using a LLM. Relevant textual sections are extracted from a corpus of scientific articles, and the CLIP model is employed to retrieve the most semantically aligned image for each section. These image–text pairs are transformed into prompts, which, through a one-shot learning approach with carefully designed instructions, guide a large language model (e.g., ChatGPT-4) to generate both detailed textual descriptions and structured triplets in the form of [Subject–Predicate–Object]. The resulting dataset follows a standardized format: [prompt (combining the section and the image), triplets], and consisted of 230 ground truth examples validated by a 4D printing expert, because direct prompting of frontier models frequently produces hallucinated or ontology-inconsistent triplets and therefore still requires extensive expert validation.

During inference, a single multimodal prompt was constructed for each target section. This prompt included: (i) the raw section text, (ii) the associated figure or schematic, and (iii) a directive stating "Extract all domain-relevant triplets". This prompt was then processed by our MLLM, which jointly attended to textual tokens and image patches to generate a set of [subject, predicate, object] assertions. For figures, the model first employed an optical character recognition (OCR) module to detect and encode text regions, and to extract key graphical elements (i.e., shapes, connectors, symbols) as visual tokens. These visual tokens interacted with text embeddings via cross-attention within the multimodal transformer. The text embeddings had been refined through our fine-tuning procedure, therefore allowing for better alignment with domain-specific semantics. This cross-modal mechanism enabled the model to infer high-level semantic relations that are not explicitly stated in the input but emerge from a combination of spatial configurations, textual cues, and prior knowledge encoded in the pretrained weights. For example, in a section from Peng et al.'s paper (Peng et al., 2022), illustrated in **Figure 3A**, describing liquid crystal elastomer (LCE) preparation, the spatial proximity and labels of "EDDET" and "PETA" enabled the model to infer an `isCrosslinkedWith` relation, which leverages both the visual structure and domain-specific patterns learned during pretraining. As shown in **Figure 3B**, relations grounded purely in the text (e.g., LCE_ink `isComposedOf` TEA) are rendered in blue, whereas those inferred from the figure's spatial layout and graphical elements (e.g., RM257 `isCombinedWith` RM82) appear in red. This example illustrates why multimodality is essential in our setting: many scientifically relevant relations are encoded exclusively within figures or schematics rather than in the surrounding text. This integrated multimodal approach thus ensures a reliable extraction of triplets from both explicit textual descriptions and implicit visual patterns.

## 2.2 Ontology Enrichment From Existing Datasets

To enhance the ontology, specific datasets that align with the domain's requirements are incorporated. The selection process considers both the relevance of the datasets and their compatibility with format constraints. Integration into the ontology follows a systematic methodology involving detailed data preparation and mapping. Each dataset is decomposed into its constituent columns, which are described and cataloged, with examples provided for clarity. To categorize each attribute within the ontology, a one-shot learning approach (Li et al., 2023; Ucar et al., 2020) supported by a large language model (Jiang et al., 2023) is applied. Each cell in every row is instantiated as an individual of its corresponding ontology class, as illustrated in **Figure 4**, and resource description framework (RDF) object properties are extracted to link these instances. For example, in the Hy-

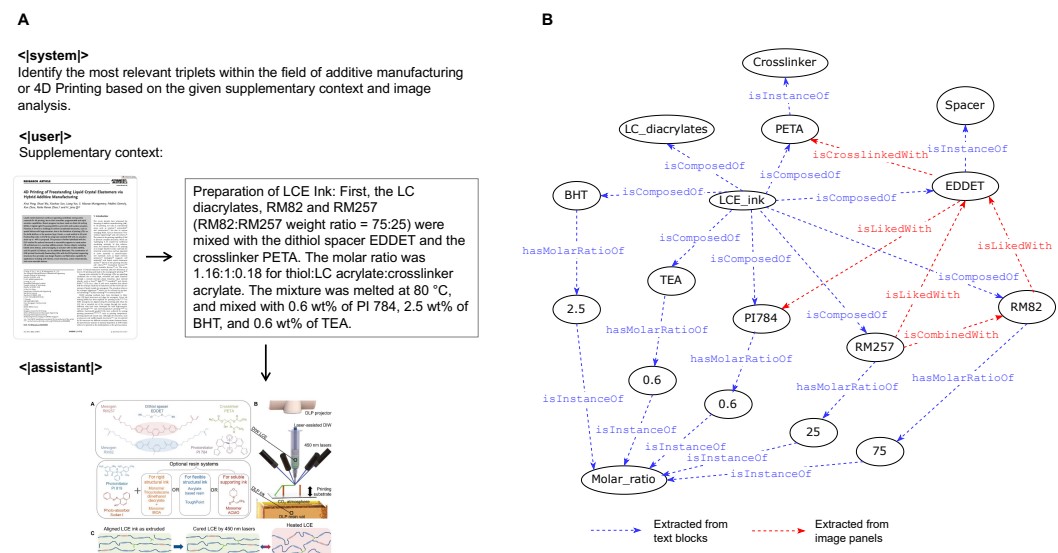

Figure 3: Example of the inference pipeline: (A) a section is selected from a published article in *Advanced Materials* Journal (Peng et al., 2022), its most relevant image was retrieved, and (B) a corresponding graph of triplets was produced.

drogel Design dataset (hyd, 2023; Richbourg et al., 2021; Richbourg & Peppas, 2023) we created instances for classes such as Polymer, SwellingRatio, and ShearModulus_kPa, and generate relations – e.g., Polymer linked via `hasShearModulus` to ShearModulus_kPa, Polymer linked via `hasSwellingRatio` to SwellingRatio, and SwellingRatio linked via `influencesShearModulus` to ShearModulus_kPa. This end-to-end pipeline yields a richly interconnected ontology graph that faithfully captures both the structural typology and the relational semantics of the original data.

## 2.3 ONTOLOGY ENRICHMENT FROM KNOWLEDGE GRAPH

Furthermore, large-scale domain knowledge graphs can be leveraged to enrich ontologies with structured knowledge. Their integration typically relies on a systematic transformation pipeline that represents information in the standard [Subject, Relation, Object] format. To ensure semantic consistency and interoperability, relationship mapping strategies are applied to align the extracted relations with the target ontology.

## 2.4 PREPROCESSING AND CONSTRUCTION OF THE ONTOLOGY

The results produced by the framework are subjected to a rigorous cleaning process to ensure that only high-quality triplets are retained. In particular, the evaluation process considers the following aspects :

**Domain relevance:** Each triplet's subject and object are transformed into contextualized embeddings using BERT and compared against embeddings derived from a curated list of domain-specific keywords. For each triplet element, the framework computes cosine similarity scores against all domain keywords and retains the maximum similarity value as the relevance indicator. The final domain relevance assessment combines both subject and object relevance scores in the overall evaluation function. This process ensures that the data is deeply aligned with the target field.

**Semantic coherence:** The framework implements a comprehensive evaluation strategy to assess semantic meaningfulness. It computes direct BERT-based cosine similarity between subject and object embeddings to measure their semantic relatedness. The final coherence score integrates both the relation validity and subject-object similarity components. Additionally, predicate coherence is evaluated through template-based assessment, where the framework compares BERT embeddings

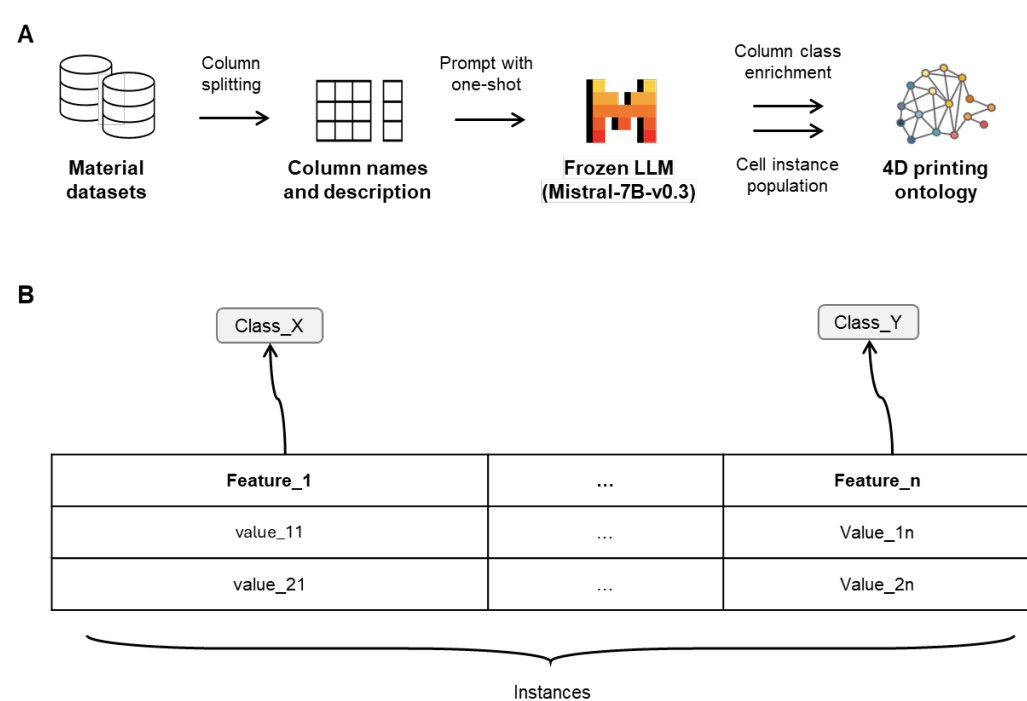

Figure 4: (A) Overview of the dataset pipeline, and (B) illustration of identified classes and intances related to a dataset representation (adapted from (Bougzime et al., 2025b)).

of complete triplet phrases against baseline phrases and relationship templates to ensure predicate appropriateness within the semantic context.

**Structural validity:** The framework checks the syntactic correctness of each triplet by verifying that all elements (subject, predicate, and object) are present, of sufficient length, and follow expected formatting standards. This validation ensures data reliability for downstream applications.

**Redundancy elimination:** Duplicate or highly similar triplets are identified through a two-stage process. First, exact duplicates are removed through string matching of subject-predicate-object combinations. Second, semantic duplicates are detected by computing BERT-based cosine similarity between triplet embeddings, where triplets exceeding a similarity threshold are flagged as redundant. This ensures that the final dataset is concise and free from both literal and semantic redundancy.

Together, these validation steps contribute to a robust and high-fidelity cleaning process that prepares the data for subsequent ontology construction and analysis. In addition to these quality control measures, the ontology construction phase integrated several advanced techniques to further enhance the ontology. First, entity names are normalized and cleaned to create valid uniform resource identifier fragments, thereby ensuring semantic consistency across the ontology. This preprocessing step effectively mitigates errors arising from formatting discrepancies or lexical variations. Furthermore, the framework incorporates a BERT-based similarity analysis that compares new class labels with those already present in the ontology. This mechanism dynamically identifies semantically similar classes and, when a sufficient similarity threshold is met, establishes subclass relationships. In doing so, the ontology consolidates redundant entities and organizes them hierarchically in a manner that mirrors the underlying domain structure. Moreover, special attention has been given to maintaining the homogeneity of the complete ontology by enforcing uniform naming conventions and consistent semantic representations across all entities. This ensures that the entire knowledge base exhibits a high degree of internal consistency, which is critical for efficient reasoning and data integration.

# 3 RESULTS: APPLYING THE FRAMEWORK TO 4D PRINTING ONTOLOGY

The rapid advancements in 4D printing have introduced a need for a structured framework to manage and formalize the diverse knowledge involved in designing transformable systems. The HERMES ontology addresses this need by providing a semantic and logical foundation for representing the dynamic behavior of 4D-printed objects (Dimassi et al., 2021). Built upon the Basic Formal Ontology (Arp et al., 2015) and mereotopology theory (Smith, 1996), this ontology is centered on key 4D printing views, namely AM, material, transformation process, and design and engineering. Although structured around philosophical foundations and DL rules to ensure expressivity and reasoning across abstraction levels, this ontology – like most existing material ontologies – suffers from limited capabilities for automated and large-scale learning through enrichment and population. This limitation is particularly critical in emerging and rapidly evolving research domains like 4D printing, where knowledge consolidation is essential to enhance technological readiness levels and reach practical applications.

To enrich the ontology, the process starts with the selection of key terms, ie., "Additive Manufacturing", "3D/4D Printing", "Shape Memory Polymer", "Shape Memory Alloy", "Liquid Crystal Elastomer", "Hydrogel", "Active/Smart Material", "Metamaterial", and "Multi-Material Structure". By identifying the pertinent intersections among these keywords, more than 1,810 relevant publications were retrieved. These articles are then decomposed into textual sections and extracted figures, which are encoded into dense vectors and indexed within a high-performance retrieval store. In parallel, material datasets collected from eight specialized databases (Jain et al., 2013b; Kuenneth & Ramprasad, 2022; hyd, 2023; Crews et al., 2012; University of Chicago, 2023; Jain et al., 2013a; Takahashi et al., 2024; NASA, 2025) undergo a column-centric processing pipeline: column names and descriptions are parsed and mapped to ontology classes using a one-shot prompting technique with an LLM, thereby instantiating each row as an instance of its corresponding class and uncovering relationships among the fields. At inference, the MLLM retrieves the most relevant text or image snippets and generates context-aware outputs, from which a dedicated extraction module derives candidate triples. These newly extracted triples, together with pre-existing entries from the MATKG knowledge graph (Venugopal & Olivetti, 2024), are then passed to a downstream symbolic reasoner. The reasoner performs rigorous validation—ensuring coherence, semantic consistency, structural integrity, and duplicate elimination—before constructing and enriching the HERMES ontology. The quality of the extracted triplets is underpinned by a Graph BERTScore F1 (Saha et al., 2021) of 0.7, demonstrating high semantic fidelity (see Appendix A). This integrated multimodal approach thus ensures a reliable extraction of triplets from both explicit textual descriptions and implicit visual patterns.

Our framework initiates the ontology enrichment process with an initial 4D printing ontology, which comprises only 170 classes, 9 instances, 48 object properties, and 13 data properties. Through the successive integration of heterogeneous data sources and advanced validation techniques, the framework has dramatically enriched and populated the ontology. In the first phase, the system processed a corpus of scientific articles by extracting triplets that describe various domain-specific relationships. In total, approximately 130,000 triplets were initially generated from articles, of which only about 28,000 were retained after applying the full consistency filtering pipeline, including semantic similarity checks, predicate template matching, and redundancy elimination. This stage resulted in the identification of 5,706 classes, 16,651 instances, 1,331 object properties, 4,390 data properties, and the establishment of 7,913 subclass relationships. The consideration of MatKG further augmented the ontology by processing additional instance-of relationships. It was responsible for incorporating 6,629 new instances and two additional data properties with 445,370 relations. This considerable increase reflects the framework's ability to integrate detailed instance-level data from supplementary sources, thereby enhancing the granularity and applicability of the ontology. A further enrichment occurred through the automated ingestion of multiple datasets from an external directory. This step contributed 144 additional classes, 12,540,671 instances, 26 object properties, and 113 subclass relationships (see **Figure 5**). By parsing and merging these large-scale datasets, the framework ensured a comprehensive and diverse coverage of the domain knowledge, while maintaining structural validity and eliminating redundancy.

After synthesizing the contributions from the scientific literature, the MatKG module, and additional datasets, the final ontology exhibits 5,849 classes, 12,563,951 instances, 1,357 object properties, 4,392 data properties, and 8,196 subclass relationships. This substantial ontology expansion

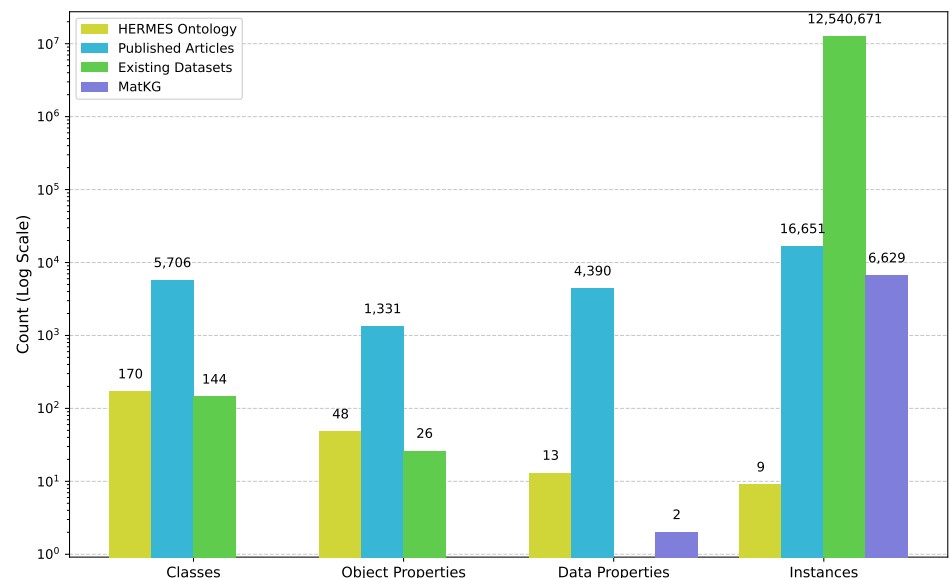

Figure 5: Comparison of ontology components between the baseline HERMES ontology and its extended counterparts derived from published articles processing, dataset parsing, and integration with MatKG triplet (adapted from (Bougzime et al., 2025b)).

demonstrates the efficacy of our multi-stage enrichment process, as illustrated in Appendix D, which highlights a subgraph centered on the stimuli-sensitive Hydrogel class.

In summary, the integration of multiple data sources, coupled with advanced NLP and robust validation measures, has culminated in a high-fidelity, richly structured ontology. Although our experimental validation focuses on 4D printing, this domain is inherently cross-disciplinary, spanning materials science, chemistry, additive manufacturing, process engineering, and smart structures. This framework is entirely domain-agnostic: adapting it to a new field would primarily involve changing the input corpus and the target ontology, while leaving the core retrieval, extraction, and validation pipeline unchanged. The resulting ontology not only represents a substantial expansion in scale and detail compared to its initial state but also provides a solid foundation for downstream applications such as knowledge-based reasoning, data integration, and semantic information retrieval across complex scientific and technical domains. When embedded within a neuro-symbolic AI (NSAI) framework, the ontology can be dynamically updated in real-time and reasoned over alongside neural models, thereby bridging symbolic and neural approaches for a context-aware design strategy (Bougzime et al., 2025a).

## 4 QUERY–ANSWER EXAMPLES FOR SMART MATERIALS AND STRUCTURES

The enriched ontology enables the formulation and resolution of design-oriented queries in 4D printing. To demonstrate its relevance, we encoded a material selection need as a semantic query vector in the ontology embedding space and retrieved the most relevant materials, experiments, and architectures by ranking their embeddings using cosine similarity. In this illustrative case, the design intent was: "Identify a soft, biocompatible material that is chemically compatible with LCE and printable via direct ink writing (DIW)".

The highest-ranked neighborhood returned by cosine similarity was dominated by silicone-based elastomers. This ranking was supported by convergent evidence from the literature, including DIW-processable silicones such as polydimethylsiloxane (PDMS) formulations.

As the top-ranked result, the ontology retrieved PDMS as a soft silicone elastomer that can be processed by DIW using embedded/freeform strategies, where uncured PDMS inks were extruded into a hydrophilic support bath and subsequently cured into stable 3D features (Hinton et al., 2015; Li & Li, 2022). PDMS is also widely documented as biocompatible and chemically inert, which explains its routine use in soft implantable systems (McDonald & Whitesides, 2002). Crucially for LCE compatibility, PDMS has been used as the passive silicone layer in bilayer systems integrating cholesteric or nematic LCE actuators, demonstrating that silicone-LCE interfaces can be formed and operated without reported interfacial chemical inhibition. Jiang et al. (2024); Li et al. (2024) Representative extracted triples include [PDMS, isPrintedUsing, DIW], [PDMS, is, Biocompatible], and [LCE actuator, isIntegratedWith, PDMS layer], confirming PDMS as a DIW-printable, biocompatible material compatible with LCE systems.

We then evaluated a second design need: "Determine the material distribution of LCE and PDMS to ensure actuation and biocompatibility of the structure". The ontology returned an embedded composite architecture, in which discrete LCE particles are dispersed within a continuous PDMS elastomer matrix. This configuration corresponds to reported polymer-dispersed LCE composites, where LCE particles provide localized anisotropic actuation and the PDMS matrix serves as a compliant host that transmits deformation while avoiding mechanical clamping of the LCE domains. Representative triples supporting this match include [LCE particles, isDispersedIn, PDMS matrix], [LCE composite, exhibits, Reversible actuation], and [PDMS, is, Biocompatible], all grounded in reported studies on LCE–PDMS composites and PDMS-based biomedical systems (Bobnar et al., 2023; McDonald & Whitesides, 2002).

The enriched ontology demonstrated strong potential as a design-support system for 4D printing by enabling high-level, intent-driven queries to be translated into relevant recommendations. By embedding design intents into a semantic vector space and retrieving knowledge via similarity ranking, the ontology accurately identified DIW-printable, biocompatible silicone elastomers – particularly PDMS – as optimal matches for a active, LCE-compatible material, with results supported by literature-derived triples and experimental reports. It further inferred an appropriate LCE–PDMS composite architecture, retrieving evidence for embedded LCE domains within a compliant PDMS matrix that preserves actuation while ensuring biocompatibility. These examples illustrate the ontology's ability to integrate heterogeneous knowledge, infer mechanistically coherent solutions, and guide material and structural design decisions, highlighting its promise as a generalizable and evidence-backed tool for accelerating discovery in smart material systems.

## 5 CONCLUSION

In this work, we presented an innovative framework for ontology enrichment applicable across diverse domains, integrating MLLMs and RAG to overcome the limitations of traditional ontological systems. Our approach, successfully combines the formal rigor of structured knowledge representation with the adaptive and contextual capabilities of advanced language models, which systematically captures heterogeneous information from scientific literature, databases and extensive knowledge graphs. Experimental results demonstrate that our methodology significantly expanded an initial, rather limited ontology – starting from 170 classes and a few instances – to a comprehensive structure encompassing over 5,800 classes and more than 12.5 million instances. Future work should focus on (i) designing specialized agent architectures that integrate vision encoders and domain-specific prompt templates for materials science modalities (Bougzime et al., 2025c;d), (ii) implementing advanced verification heuristics that leverage both linguistic and visual ontological rules, (iii) developing evaluation metrics for multimodal triplet extraction that reflect the unique challenges of materials knowledge representation, and (iv) creating dedicated relation classification agents for precise typing along with specialized validation agents for ontology cohesion and triplet integrity. By embracing multimodal multi-agent systems, we can move toward adaptive ontologies that evolve seamlessly with the scientific literature, providing researchers with powerful tools for accelerated materials discovery and development.

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

## A  AI Model Assessment

To determine the optimal temperature – a key parameter that regulates the level of randomness in the model's output during inference – we evaluated the model's performance across a range of temperature settings. Specifically, we tested two configurations: the fine-tuned model on its own, and the fine-tuned model combined with one-shot learning. Temperature plays an important role in balancing determinism and creativity in language model outputs. Lower temperatures make the model's responses more focused and predictable, while higher temperatures increase variability and originality. This trade-off impacts the accuracy and relevance of extracted triplets (Murel & Noble, 2024). As illustrated in **Figure 6**, the standard fine-tuning approach without any in-context learning demonstrated higher stability and improved performance when compared to the fine-tuning approach with one-shot across metrics which represent n-gram-based metrics encompassing precision (Bilingual Evaluation Understudy, termed as BLEU), Recall-Oriented Understudy for Gisting Evaluation (termed as ROUGE), and F1-score (combining BLEU and ROUGE metrics) (Ghanem & Cruz, 2024). These n-gram-based metrics rely on the comparison of overlapping word sequences (called n-grams) between the generated and reference texts. For instance, an $e$-gram refers to a contiguous sequence of $e$ words, 1-grams are unigrams (single words), 2-grams are bigrams, and so on, thus providing nuanced evaluation of fluency and relevance in generated text (Jurafsky & Martin, 2025). Details of the metric computation are provided in the next section.

In addition to large-scale quantitative enrichment, we performed a qualitative assessment by manually reviewing a representative subset of extracted triples (top-k highest confidence predictions and a random sample of 200 instances) compared to expert knowledge. The evaluation considered four dimensions: (i) relevance of the extracted relation to the 4D printing domain, (ii) factual correctness with respect to the source text, (iii) clarity of entity boundaries and relation semantics, and (iv) actionability in terms of whether the triple can be meaningfully integrated into downstream ontology reasoning. For instance, the extracted triple [Hydrogel, `isInstanceOf`, Biocompatible_Material] was judged as fully correct and relevant, whereas [3D_printing, `converts`, Robust_manufacturing_process], although syntactically valid, was marked as semantically unclear and thus of limited actionability. Overall, 84% of the reviewed triples were rated as both factually correct and relevant, 10% as partially correct but ambiguous (e.g., inconsistent entity typing), and 6% as incorrect due to model hallucinations or misalignment. This analysis highlights that while large-scale automatic enrichment can produce "shallow" or noisy entries, systematic sampling and expert review confirm that the majority of extracted triples are suitable for ontology integration. Qualitatively, these proportions indicate that high-confidence extractions are immediately usable, whereas most residual errors stem from entity typing and boundary delineation; this highlights schema-aware normalization and predicate templating as the primary levers for improvement. Error forensics further show that low-evidence contexts and predicate drift account for the majority of the remaining 6% failures, motivating a quarantine queue and tighter domain/range constraints prior to graph insertion. Taken together, these observations align with our objectives—precision under ontological constraints and provenance-backed integration—and directly inform corrective actions in the pipeline (stricter typing, evidence thresholds, and conflict-aware deduplication). Beyond this qualitative view, we performed a quantitative analysis showing that the stand-alone fine-tuned model consistently outperforms the fine-tuned-plus-one-shot configuration, which exhibits pronounced variability and uniformly lower scores. Triplet-matching F1 peaks at T≈0.55, while G-BLEU and G-ROUGE remain optimal over T∈ [0.55, 0.70]; G-BERTScore precision is maximal near T≈0.55, indicating fine-grained semantic alignment between predicted and reference graphs. Taken together, these results show that T≈0.55 offers the best precision–recall trade-off, combining robust F1 with sensitivity captured by both n-gram overlap and contextualized embeddings. Consequently, plain fine-tuning delivers superior extractive accuracy and stability for relational triple extraction while avoiding the complexity and instability introduced by one-shot in-context learning. Analytically, this justifies fixing T= 0.55 as the default decoding regime to prioritize precision without collapsing recall; operationally, it implies fewer low-confidence triples entering pre-insertion quarantine and, downstream, fewer ontology-coherence violations – i.e., a larger yet cleaner graph.

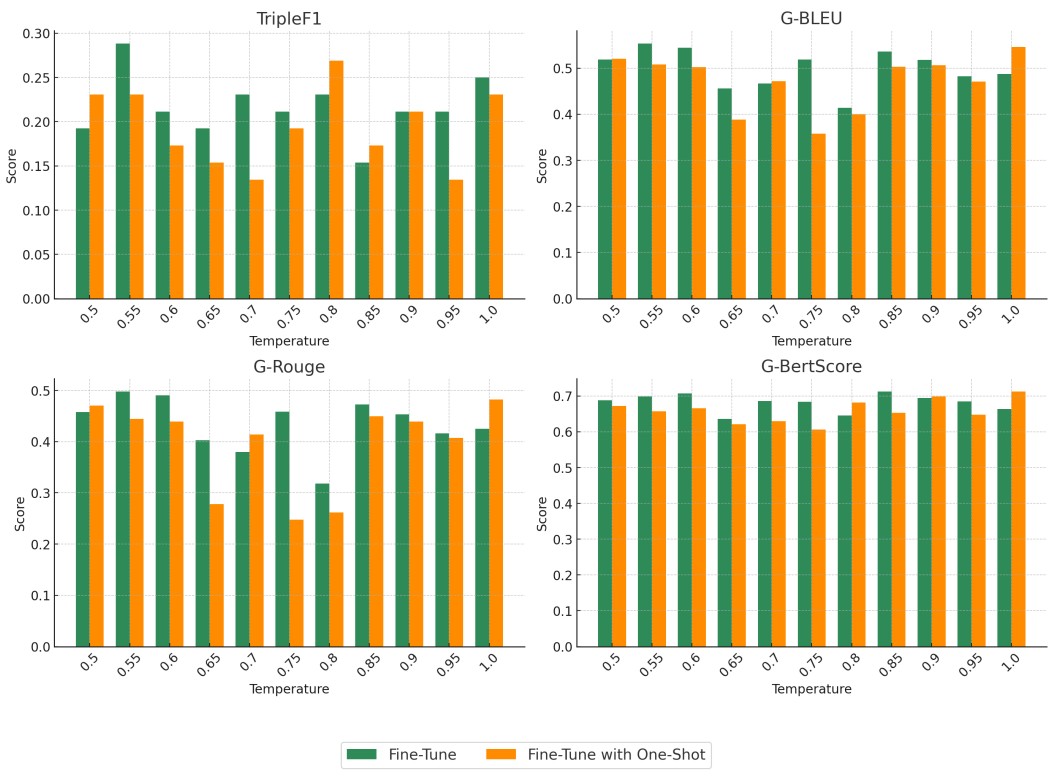

Figure 6: Comparison metrics between prompt with the fine-tuned model vs. prompt with the fine-tuned model using one-shot technique performance across various temperatures (adapted from (Bougzime et al., 2025b)).

## B  METRIC COMPUTATION

**BLEU-F1 Score (Papineni et al., 2002)**

Let $C_{\text{gen}}$ be the number of 4-grams in the generated graph, $C_{\text{ref}}$ the number of 4-grams in the reference graph, and $C_{\text{match}}$ the number of matching 4-grams. Then:

$$P_{\text{Bleu}} = \frac{C_{\text{match}}}{C_{\text{gen}}} \tag{1}$$

$$R_{\text{Bleu}} = \frac{C_{\text{match}}}{C_{\text{ref}}} \tag{2}$$

$$F1^{\text{Bleu}} = \frac{2\,P_{\text{Bleu}}\,R_{\text{Bleu}}}{P_{\text{Bleu}} + R_{\text{Bleu}}} \tag{3}$$

**ROUGE-F1 Score (Lin, 2004)**

For ROUGE-2 (bigrams), let $bigram_{\text{cand}}$ and $bigram_{\text{ref}}$ be the sets of bigrams in the candidate and reference, respectively. Then

$$P_{\text{ROUGE}} = \frac{|bigram_{\text{cand}} \cap bigram_{\text{ref}}|}{|bigram_{\text{cand}}|} \tag{4}$$

$$R_{\text{ROUGE}} = \frac{|bigram_{\text{cand}} \cap bigram_{\text{ref}}|}{|bigram_{\text{ref}}|} \tag{5}$$

$$F1^{\text{ROUGE}} = \frac{2\,P_{\text{ROUGE}}\,R_{\text{ROUGE}}}{P_{\text{ROUGE}} + R_{\text{ROUGE}}} \tag{6}$$

**Graph BERTScore (G-BS) (Saha et al., 2021)**

G-BS takes graphs as a set of edges and solves a matching problem which finds the best alignment between the edges in the predicted graph and those in the ground-truth graph. Each edge is considered as a "sentence" and BERTScore is used to calculate the similarity between a pair of predicted and ground-truth edges. Based on the optimal alignment and the overall matching score, the computed F1 score is used as the final G-BERTScore.

Considering $x_i$ as a reference token (entity or relation) and $\hat{x}_j$ as a generated token (entity or relation), the complete score matches each token in $x$ to a generated token in $\hat{x}$ to compute recall, and each token in $\hat{x}$ to a token in $x$ to compute precision. A greedy matching was used to maximize the total similarity score, where each token was matched to the most similar token in the other graph. Then precision and recall were combined to compute an F1 measure.

For a reference sequence $x = \{x_1, \ldots, x_{|x|}\}$ and a candidate sequence $\hat{x} = \{\hat{x}_1, \ldots, \hat{x}_{|\hat{x}|}\}$, the recall $R$, precision $P$, and F1 scores are defined as:

$$R_{\text{BERT}} = \frac{1}{|x|} \sum_{x_i \in x} \max_{\hat{x}_j \in \hat{x}} x_i^\top \hat{x}_j \tag{7}$$

$$P_{\text{BERT}} = \frac{1}{|\hat{x}|} \sum_{\hat{x}_j \in \hat{x}} \max_{x_i \in x} x_i^\top \hat{x}_j \tag{8}$$

$$F_1^{\text{BERT}} = \frac{2\, P_{\text{BERT}}\, R_{\text{BERT}}}{P_{\text{BERT}} + R_{\text{BERT}}} \tag{9}$$

**Triples Matching (T-F1)**

The F1 score for triple matches T-F1 was calculated as follows:

$$T\text{-}F1 \;=\; \frac{2\,\text{TP}}{2\,\text{TP} + \text{FP} + \text{FN}} \tag{10}$$

where TP is the number of true positive triple matches, FP is the number of false positive triple matches, and FN is the number of false negative triple matches. These metrics could potentially yield even better results if synonyms of entities or relations are considered as exact matches. This evaluation serves as a benchmark analysis for assessing the efficacy of our ontology enrichment methodology.

## C  EXPERIMENTAL SECTION

LLaVA was fine tuned using LoRA to improve triplet extraction. This technique allowed us to efficiently adjust the model's parameters while minimizing resource consumption. The fine-tuning process utilized a synthetic dataset specifically designed to include textual and visual contexts aligned with the needs of 4D printing. In this process, LoRA was enabled with a rank of 128 and an alpha value of 256, and the multimodal projector learning rate was set to $2 \times 10^{-5}$. The base model used was liuhaotian/llava-v1.5-7b (version v1), and the training data comprised a synthetic dataset alongside an image folder containing the extracted images. The vision tower was configured to use OpenAI's CLIP ViT-Large-Patch14-336, while the multimodal projector was set to an MLP with GELU activation (mlp2x_gelu), selecting the penultimate layer for vision features. Notably, both image start-end tokens and image patch tokens were disabled, and images were preprocessed with a padded aspect ratio. The training procedure grouped samples by modality length and leveraged bfloat16 precision. Training was conducted for 5 epochs with a per-device batch size of 8 for training (and 4 for evaluation), accumulating gradients over 2 steps. The evaluation strategy was disabled during training, and model checkpoints were saved every 50,000 steps with a total checkpoint limit of one. The learning rate was fixed at $2 \times 10^{-4}$ with no weight decay and a warmup ratio of 3%, using a cosine learning rate scheduler. Additionally, TF32 was enabled, the maximum model length was set to 2,048 tokens, gradient checkpointing was employed to conserve memory, and 4 dataloader workers were used in conjunction with lazy preprocessing. **Figure 7** shows a rapid early loss drop followed by a low-variance plateau ($\sim$0.05), indicating stable convergence under LoRA (rank= 128, $\alpha = 256$) and supporting this fine-tuned LLaVA configuration as an appropriate and reliable model for triplet extraction.

All inferences were run on a single NVIDIA A40 GPU, with an average runtime of about one minute per prompt for the fine-tuned model, compared to approximately seven minutes per prompt for the few-shot configuration, highlighting a substantial gain in computational efficiency.

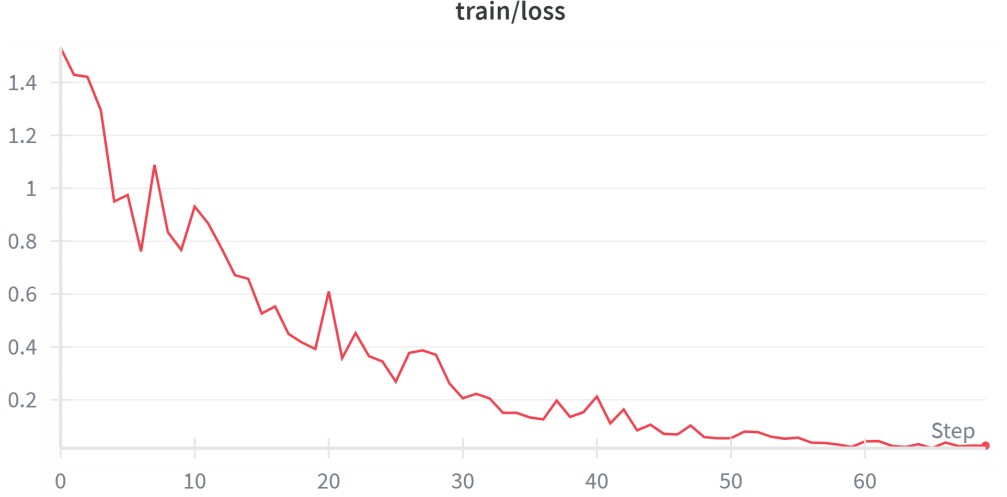

Figure 7: Training loss during LoRA fine-tuning of LLaVA for triplet extraction (reported by Weights & Biases).

## D ONTOLOGY VIEW CENTERED ON THE HYDROGEL CLASS

This appendix provides a zoomed-in view of the final enriched ontology around the **Hydrogel** class. The graph illustrates how literature-derived triplets, dataset instances, and knowledge graph imports jointly populate stimuli-sensitive hydrogels and their related properties, processes, and functional roles (See **Figure 8**).

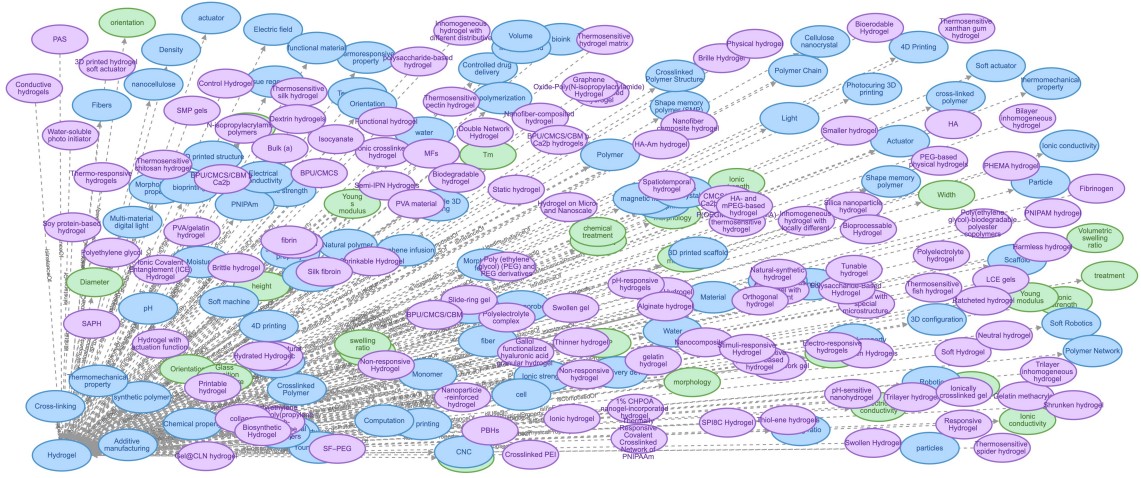

Figure 8: Final enriched 4D printing ontology through the lens of the stimuli-sensitive Hydrogel class. To ensure clarity, only a limited number of classes, object properties, and instances are displayed (adapted from (Bougzime et al., 2025b)).

