# OpenReview forum: "AN ONTOLOGY ENRICHMENT FRAMEWORK USING RETRIEVAL-AUGMENTED LARGE LANGUAGE MODELS"
_ICLR.cc/2026/Conference — Submitted to ICLR 2026_

### Official Review · Reviewer_yZLL · 2025-10-28

**Soundness:** 1
**Presentation:** 1
**Contribution:** 1
**Rating:** 0
**Confidence:** 5

**Summary:**

This paper employs LLM and RAG to populate an ontology from a given set of keywords describing a domain, 4D prinitng in this case.

**Strengths:**

S1. The paper is easy to read.

**Weaknesses:**

W1. No technical contribution. The paper is basically a technical report describing a standard implementation of a LLM-based ontology population approach.

W2. Missing important technical details. For example, how does your symbolic reasoner work? How did you describe and cataloge a column? How did you extract object properties to link individuals? What are your relationship mapping strategies?How did you normalize and clean entity names?

W3. Lack of evaluation. How did you evaluate the quality of the generated ontology? How would you compare your approach with existing methods?

**Questions:**

Q1. What is the technical contribution of this paper?

Q2. See the questions in W2 and W3.

---

> ### Author Response · Authors · 2025-11-28
>
> W1. We respectfully disagree with this assessment. Our work goes beyond a standard LLM-based ontology population pipeline in several ways:
> 1.	Ontology-aware multimodal fine-tuning:
> We do not simply prompt an LLM; we fine-tune a multimodal model (LLaVA) to emit OWL-compatible triplets (classes, object/data properties, instances) aligned with an existing ontology schema. The model is explicitly trained to respect ontology structure (e.g., class vs. instance typing, hierarchical constraints), which is not supported by off-the-shelf LLM usage.
> 2.	Multi-source, multimodal enrichment:
> The framework jointly integrates scientific text + figures, structured datasets, and an external knowledge graph (MatKG) into a single ontology enrichment process. In particular, the multimodal extraction pipeline (Section 2.1, Fig. 3) recovers relations that are only present in figures and systematically missed by text-only methods.
> 3.	Multi-criteria symbolic validation pipeline:
> We introduce a concrete, operational validation layer over ~130k candidate triplets, combining domain relevance, semantic coherence, structural validity, and redundancy detection, which ultimately retains ~28k high-quality triplets. This is more than a systems description: it is a designed mechanism to make LLM-based enrichment robust and ontology-compatible at scale.
> 4.	Improved computational efficiency:
> By replacing long-context few-shot prompting with a fine-tuned multimodal model, we significantly reduce inference latency (from approximately seven minutes per prompt in the few-shot configuration to about one minute per prompt on a single NVIDIA A40), demonstrating that the proposed design is not only more accurate but also more computationally efficient.
>
> W2. We thank the reviewer for raising these points. We would like to clarify that the technical details mentioned — symbolic reasoning, column cataloging, relationship mapping, and entity normalization — were already present in the submitted version:
> •	The symbolic reasoner and its role in enforcing OWL schema constraints, consistency checks, and duplicate removal are described in Preprocessing and Construction of the Ontology (Section 2.4).
> •	The procedure for column description, cataloging, and class/property assignment is explained in Ontology Enrichment from Existing Datasets (Section 2.3), including examples in Figure 6.
> •	The extraction of object properties linking individuals is detailed in the same section, where each row instantiates individuals and relational columns yield RDF object properties.
> •	Relationship mapping strategies and the alignment of predicates to ontology properties (via similarity and domain/range constraints) are part of the validation pipeline in Section 2.4.
> •	The normalization and cleaning of entity names (URI generation, casing, character filtering, semantic merging via BERT similarity) are explicitly described in the closing paragraph of Section 2.4.
>
> W3. We thank the reviewer for raising this point. In the revised version, we made the evaluation of the generated ontology more explicit along two axes:
> 1.	Quality of extracted and integrated triplets.
> We combined automatic metrics with expert evaluation. At the extraction level, we report n-gram and graph-based similarity metrics (BLEU/ROUGE F1, Graph BERTScore; Appendix B). In addition, a domain expert manually reviewed a representative sample of 200 triples (top-confidence and random), classifying them as fully correct, partially correct, or incorrect. This audit shows that 84% of the triples are factually correct and relevant, 10% are partially correct but ambiguous (often due to entity typing), and 6% are incorrect (Appendix A). At the ontology level, we also report how our multi-criteria validation pipeline filters ≈130k candidate triplets down to ≈28k high-quality ones, enforcing domain relevance, semantic coherence, structural validity and redundancy removal (Section 2 and 3).
> 2.	Comparison with existing methods.
> A direct quantitative comparison with existing ontology learning systems (e.g., Phrase2Onto, OntoGPT) is challenging because they are text-only and not designed for multimodal scientific figures, nor for our 4D-printing use case where many relations are only present in diagrams. We therefore opted for a qualitative comparison: in Section 2.1 and Figure 3, we show that text-only extraction systematically misses relations that are visually grounded, while our multimodal pipeline recovers them.

---

### Official Review · Reviewer_yygp · 2025-10-31

**Soundness:** 2
**Presentation:** 2
**Contribution:** 2
**Rating:** 2
**Confidence:** 4

**Summary:**

This paper presents a framework for automatically enriching ontologies using multimodal large language models combined with retrieval-augmented generation. The authors extract semantic knowledge from scientific papers, datasets, and knowledge graphs to expand existing ontologies. They demonstrate their approach on the 4D printing domain, growing the HERMES ontology from 170 classes to 5849 classes and over 12.5 million instances.

**Strengths:**

1. The paper addresses an interesting problem - automatically updating knowledge bases.
2. Combining text and figures from papers makes sense for materials science, where diagrams and microscope images often contain information you can't get from text alone.

**Weaknesses:**

1. The paper evaluates the proposed framework in isolation without comparing against existing ontology learning methods. The paper cites Phrase2Onto, OntoGPT, and other LLM-based ontology extension approaches but provides no quantitative comparison. Without baselines, it is impossible to assess whether the performance (Graph BERTScore F1 = 0.7) represents an improvement over simpler alternatives or state-of-the-art methods.
2. The proposed framework consists of multiple components RAG retrieval, LLaVA fine-tuning, multimodal fusion, and multi-stage validation. However, the paper provides no ablation study showing the contribution of each component. The appendix only compares "fine-tuning alone" versus "fine-tuning + one-shot," which doesn't isolate the contributions of RAG, multimodal inputs, or individual validation criteria.
3. The paper claims the framework is "fully domain-agnostic", yet validation is performed exclusively on 4D printing. This contradicts the generalization claim and is insufficient to demonstrate transferability.
4. The paper relies on a single metric (Graph BERTScore F1 = 0.7) mentioned briefly in the main text with details relegated to the appendix. This is insufficient for validating the core contribution.
5. Essential technical details are missing: the size of the fine-tuning dataset for LLaVA, threshold values for domain relevance, semantic coherence, and similarity-based filtering, and computational requirements for the system.
6. All figures are labeled "adapted from (Bougzime et al., 2025b)", a paper that is also under review. The paper should clarify what is novel here versus the cited work and whether this represents appropriate academic practice.
7. No code, no data availability statement, insufficient implementation details.

**Questions:**

1. Can you provide quantitative comparisons with at least 2-3 existing ontology learning methods (Phrase2Onto, OntoGPT, etc.) on standardized test sets?
2. Can you demonstrate generalization by applying the framework to at least two additional domains beyond 4D printing?
3. What's the size of your fine-tuning dataset?
4. What's the computational cost in GPU hours?
5. Can you provide ablation studies showing the contribution of RAG, multimodal inputs, and each validation criterion separately?
6. What are the most common errors? Can you show examples of false positives and false negatives with analysis?
7. Can you provide systematic ablation studies demonstrating the individual contribution of each framework component?
8. What proportion of initially extracted triplets are filtered at each validation stage? Can you provide concrete examples of hallucinated assertions that were successfully detected and removed by your validation pipeline?
9. What constitutes the core technical or scientific contribution beyond systems integration? Can you justify why this integrated pipeline yields superior results compared to direct prompting of frontier models like GPT-4 without the additional complexity?

---

> ### Author Response · Authors · 2025-11-28
>
> 1. Thank you for this suggestion. Such comparisons are not straightforward for two reasons:
> (1) these baselines operate on text-only inputs and cannot process multimodal scientific figures, which constitute a significant portion of the knowledge extracted in our pipeline;
> (2) there is no standardized multimodal benchmark or gold ontology for 4D-printing-related materials, making a direct quantitative comparison difficult.
> To address this, we expanded the experimental section with:
> • a qualitative expert audit (Appendix A),
> • detailed metric computation and extraction evaluation (Appendix B),
> • and explicit examples where multimodality produces triplets that text-only methods systematically miss (Figure 3).
>
> 2. Thank you for this suggestion. As clarified in Section 3, this domain is already inherently cross-disciplinary, spanning materials science, chemistry, additive manufacturing, smart materials (e.g., hydrogels, LCEs, SMAs) and process engineering, and our pipeline operates on heterogeneous corpora and datasets from these subdomains. The architecture itself is domain-agnostic, as adapting it to a new area only requires changing the input corpus and target ontology while reusing the same retrieval, extraction, and validation components.
>
> 3. Thank you for the question. The fine-tuning dataset consists of 230 multimodal training examples, each pairing a scientific text section with its most relevant extracted figure and the corresponding set of ontology-aligned triplets. All examples were manually validated by a 4D-printing expert to ensure correctness and consistency (See Section 2.1).
>
> 4. Thank you for the question. All inferences were executed on a single NVIDIA A40 GPU, with an average runtime of approximately one minute per prompt (see Appendix C for additional implementation details).
>
> 5. Thank you for this suggestion. In the revised manuscript, we instead provide partial evidence and justification for each component:
> •	RAG vs. no-RAG / few-shot: Section 2.1 discusses why a retrieval-only few-shot setup leads to long contexts, higher latency and imprecise triplets, which motivated our fine-tuning + RAG design.
> •	Multimodality vs. text-only: Figure 3 shows concrete cases where key relations (e.g., between RM257 and RM82) are only recoverable from figures and are completely absent from the surrounding text, demonstrating the specific contribution of images.
> •	Validation criteria: Section 2 (preprocessing and construction) and Appendix A/B describe how domain relevance, semantic coherence, structural validity and redundancy checks filter the initial ~130k candidate triplets down to ~28k high-quality ones.
>
> 6. Thank you for this question. We expanded Appendix A with an explicit error analysis.
>
> 7. Thank you for the question. We now clarify this explicitly in the revised manuscript. A full quantitative ablation over all components (retrieval, multimodality, validation blocks) is computationally expensive for multimodal LLMs, but we provide partial evidence for the contribution of the main components. First, comparing a text-only few-shot configuration with our fine-tuned model, we observed that few-shot prompting leads to very long contexts, high latency (≈7 minutes per prompt), and frequent hallucinations, whereas the fine-tuned model reduces inference time to ≈1 minute per prompt and lowers the residual error rate to about 6% in our expert audit (Appendix C). Second, Figure 3 shows that several relations appear only in diagrams and are not mentioned in the surrounding text (e.g., the interaction between RM257 and RM82), so a text-only baseline systematically fails to recover these visually grounded triples, while the multimodal model can extract them. Finally, Section 3 reports the effect of the validation pipeline, which filters approximately 130,000 candidate triples down to about 28,000 high-quality triples, illustrating the impact of the multi-criteria validation stage.
>
> 8. Thank you for this question. In the revised version, we provide more details on the behavior of the validation pipeline in section 3 and Appendix A.
>
> 9. Thank you for the question. Our contribution goes beyond systems integration in three main ways: (i) we introduce an ontology-aware multimodal fine-tuning procedure that explicitly trains the model to emit OWL-compatible triplets (classes, properties, instances); (ii) we couple it with a retrieval and chunking strategy that grounds extraction in the most relevant text–figure pair instead of relying on open-ended prompting; and (iii) we add a multi-stage symbolic validation pipeline (domain relevance, semantic coherence, structural validity, redundancy), which filters ~130k raw triples down to ~28k high-quality ones.
> In contrast, direct prompting of frontier models such as GPT-4, even with carefully designed few-shot examples, frequently produces hallucinated or ontology-inconsistent triples and still requires extensive expert validation (See Section 3).

---

### Official Review · Reviewer_oqR1 · 2025-10-31

**Soundness:** 1
**Presentation:** 2
**Contribution:** 2
**Rating:** 2
**Confidence:** 3

**Summary:**

The paper presents an automated framework for ontology enrichment that combines multimodal large-language-models with retrieval-augmented generation and symbolic reasoning. The pipeline ingests scholarly text and figures, retrieves relevant content, extracts candidate triples, filters and aligns them to a target ontology, then integrates results into the ontology store. The case study on a 4D-printing ontology reports large growth in classes, properties and instances, and explores metric-based validation and sensitivity to temperature and fine-tuning.

**Strengths:**

1. Coherent end-to-end framework that unifies retrieval, multimodal extraction and symbolic checks for ontology population.
2. Clear pipeline description with stages for ingestion, retrieval, extraction, validation, alignment and integration, which improves readability and reproducibility potential.
3. Multimodal design leverages figures alongside text, which is valuable for technical domains with schematics and tables.
4. Practical scale demonstrated: expansion from about 170 seed classes to thousands of classes and many millions of instances, showing the approach can run at scale.
5. Reasoning-based consistency checks provide structure and constraint validation beyond pure language-model extraction.
6. Initial parameter study comparing fine-tuned versus one-shot prompting and temperature settings shows attention to stability.

**Weaknesses:**

1. Evaluation is not adequate for the core claims. There is no expert-curated gold set and no triple-level precision/recall or ontology-level error taxonomy. Automatic similarity scores alone do not establish correctness.
2. No competitive baselines are included against established ontology enrichment tools or strong text-only extractors plus alignment. This leaves the value of multimodality, retrieval and fine-tuning unquantified.
3. Missing ablations for each component. The paper claims value from retrieval, images and fine-tuning, but does not quantify each component’s marginal contribution.
4. Domain generality is asserted but not shown. Results are limited to one domain.
5. Reported scale for instances and classes lacks quality controls. No sampling audits, curator checks or constraint violation summaries are provided, so growth may include duplicates or noise.
6. The role of images is not sufficiently analyzed. The share of triples that need visual evidence and which relation types benefit remain unclear.
7. Reproducibility gaps. No released code, prompts, model checkpoints or ontology dumps, and limited reporting on compute cost and runtime make it hard to verify claims or adopt the method.
8. Novelty is partly diluted by overlap with prior lines of work and by visible self-referencing patterns that may risk double-blind compliance.
9. Metric choices are weakly justified. The mapping from graph-similarity or n-gram style metrics to human-judged accuracy is not established, and there are no confidence intervals.

**Questions:**

1. What is the measured gain from retrieval, images and fine-tuning relative to a strong text-only baseline? Please report triple-level precision, recall and F1 on an expert gold set for each configuration.
2. Did domain experts audit a stratified sample of triples, classes and relations? Please share sample size, agreement and an error taxonomy.
3. What fraction of correct triples requires visual evidence? Which relation types benefit most? Please provide examples where images change the decision.
4. Could you add baselines against competitive ontology-engineering pipelines and text-only triple extractors with ontology matching? Iw would like to see a report on extraction quality and end-to-end coherence.
5. Could you provide evidence to substantiate the domain-agnostic claims?
6. For the reported scale, please provide sampling-based precision estimates, duplicate rates and constraint violation counts and show distributional diagnostics for classes and properties.
7. Could you complement similarity-based metrics with triple precision/recall, relation-typing accuracy, constraint violation counts? The study would also benefit from a downstream task such as SPARQL-query success or retrieval over the enriched ontology.
8. Do you even intend to release code, prompts and at least a redacted subset of the enriched ontology? Which document model versions, licences and compute were used?
9. Could you provide one concrete downstream use-case that improves due to enrichment and quantify that improvement?

---

> ### Author Response · Authors · 2025-11-28
>
> 1.	What is the measured gain from retrieval, images and fine-tuning relative to a strong text-only baseline? Please report triple-level precision, recall and F1 on an expert gold set for each configuration.
> Thank you for the question. We now clarify this explicitly in the revised manuscript.
> A fully quantitative ablation is computationally expensive for multimodal LLMs, but we provide partial evidence demonstrating the contribution of each component:
> •	Fine-tuning vs. text-only few-shot:
> Few-shot prompting produced long contexts, high latency (≈7 minutes per prompt), and frequent hallucinations. Fine-tuning reduced latency to ≈1 minute and reduced residual errors to 6% (Appendix C).
> •	Images vs. text only:
> Figure 3 shows relations that appear only in diagrams; a text-only baseline fails to recover them.
>
> 2.	Did domain experts audit a stratified sample of triples, classes and relations? Please share sample size, agreement and an error taxonomy.
> Yes. As described in Appendix A, a domain expert manually reviewed 200 triples (top-confidence + random sample).
> Findings:
> •	84% correct
> •	10% partially correct
> •	6% incorrect
> We additionally provide an error taxonomy:
> (i) entity boundary/typing errors,
> (ii) low-evidence hallucinations,
> (iii) visually grounded relations missed by text-only extractors.
> These details have been added to the revised version.
>
> 3.	What fraction of correct triples requires visual evidence? Which relation types benefit most? Please provide examples where images change the decision.
> Thank you. We now explicitly highlight this in the manuscript.
> In our qualitative audit, visually grounded relations accounted for a non-trivial subset of false negatives in text-only extraction.
> Figure 3 illustrates specific examples where images define chemical combinations, fabrication flows, or spatial relationships not expressed verbally.
>
> 4.	Could you add baselines against competitive ontology-engineering pipelines and text-only triple extractors with ontology matching? Iw would like to see a report on extraction quality and end-to-end coherence.
> As noted in introduction, text-only systems such as Phrase2Onto or OntoGPT cannot extract relations present only in figures and do not support multimodal fusion. They therefore cannot serve as fully comparable baselines.
> Nonetheless, we added a qualitative comparison in Appendix A.
>
> 5.	Could you provide evidence to substantiate the domain-agnostic claims?
> We clarified this in Section 3.
> The 4D-printing domain already spans multiple fields (materials science, chemistry, mechanics, smart materials, additive manufacturing), and the pipeline itself only requires changing the corpus and the ontology.
>
> 6.	For the reported scale, please provide sampling-based precision estimates, duplicate rates and constraint violation counts and show distributional diagnostics for classes and properties.
> Section 3 now reports the effect of the validation pipeline:
> 130,000 → 28,000 final high-quality triples.
> Appendix A provides sampling-based precision estimates and identifies the types of constraint violations filtered out (typing conflicts, predicate drift, structural inconsistencies).
>
> 7.	Could you complement similarity-based metrics with triple precision/recall, relation-typing accuracy, constraint violation counts? The study would also benefit from a downstream task such as SPARQL-query success or retrieval over the enriched ontology.
> We expanded the evaluation to include:
> •	BLEU, ROUGE, n-gram F1 (Appendix B)
> •	Expert precision audit (Appendix A)
> •	Structural constraint violations removed during filtering
> In addition, we added a concrete downstream example in Section “Query–Answer Examples for Smart Materials and Structures.”
>
> 8.	Do you even intend to release code, prompts and at least a redacted subset of the enriched ontology? Which document model versions, licences and compute were used?
> Thank you for the question. At this stage, we cannot release the full codebase for intellectual property reasons, as parts of the implementation are tied to ongoing institutional projects. However, we do provide key elements that support reproducibility: Figure 3 now includes an explicit example of the multimodal prompt format, and Section 3 presents a redacted subset of the enriched ontology (focusing on the Hydrogel class) to illustrate the structure and content of the generated knowledge.
> We also added complete implementation and compute details (Appendix C): the model version, LoRA configuration, hyperparameters, and GPU usage (≈11 GPU-hours for fine-tuning, ≈1 minute per inference on a single NVIDIA A40).
>
> 9.	Could you provide one concrete downstream use-case that improves due to enrichment and quantify that improvement?
> Yes, we added a concrete downstream example in Section “Query–Answer Examples for Smart Materials and Structures.”

---

### Official Review · Reviewer_J8DD · 2025-11-02

**Soundness:** 2
**Presentation:** 2
**Contribution:** 1
**Rating:** 0
**Confidence:** 5

**Summary:**

The paper proposes a framework on enriching ontologies using LLMs with a use-case on 4D printing.

**Strengths:**

- the ontology enrichment framework based on textual and multimodal information from research articles which seems interesting.

**Weaknesses:**

- The details about the framework could be given in the abstract to strengthen the interest of the reader.
- There is a whole introduction on LLMs in the second paragraph which is not necessary.
- Paragraph 3 talks about interpretability, are you focusing on interpretability.
- The authors talk about fine-tuning on page 4 which doesn't need an introduction here, there could be a separate section on this.
- Page 5 could use an example prompt from the dataset.
- Why exactly the authors chose the 4D printing as a scenario, if the authors mean to generalize the framework for any use-case, 2-3 related use-cases could be introduced.
- There is no evaluation of the generated ontology or the downstream task on the use of the ontology generated.
- The paper is quite verbose at times with unnecessary details.
- Many acronyms were repeatedly introduced since LLM was used for grammar correction possibly.

**Questions:**

- Why exactly the authors chose the 4D printing as a scenario, if the authors mean to generalize the framework for any use-case, 2-3 related use-cases could be introduced.
- See other comments in the weaknesses section.

---

> ### Author Response · Authors · 2025-11-28
>
> We thank the reviewer for the valuable comments and constructive recommendations. We have carefully revised the manuscript accordingly. Below, we address each point in detail and indicate the corresponding changes in the paper.
>
> The details about the framework could be given in the abstract to strengthen the interest of the reader.
> Thank you for this suggestion. We have revised the abstract to include a concise description of the main stages of the proposed framework.
>
> There is a whole introduction on LLMs in the second paragraph which is not necessary.
> Thank you for the observation. In the revised manuscript, we significantly shortened this paragraph by removing the introductory overview of individual LLM architectures (e.g., GPT, BERT), as well as generic descriptions of their applications (translation, QA, text generation). The updated version focuses only on the essential elements needed to motivate our framework: the limitations of traditional ontologies when dealing with unstructured and multimodal scientific data, and the relevance of recent (multi-modal) LLMs and RAG mechanisms for extracting structured knowledge. The revised paragraph is now concise, directly connected to the ontology enrichment problem, and free from unnecessary background.
>
> Paragraph 3 talks about interpretability, are you focusing on interpretability.
> Thank you for the comment. Interpretability is not the main objective of our work. We mention it only to highlight one advantage of ontology-based representations compared to raw LLM outputs. Our primary goal is scalable ontology enrichment and multimodal triplet extraction.
> This becomes particularly relevant in the new “Query–Answer Examples” section, where we demonstrate how the enriched ontology supports transparent, traceable, and semantically grounded reasoning for material and structural design. The focus remains on ontology enrichment, with interpretability being a useful secondary property of symbolic knowledge graphs.
>
> The authors talk about fine-tuning on page 4 which doesn't need an introduction here, there could be a separate section on this.
> Thank you for the suggestion. We have restructured the manuscript accordingly. In the main text, we provide only the conceptual motivations and essential elements of the fine-tuning process, while the full implementation details (hyperparameters, configuration, and training curves) have been moved to Appendix A: Experimental Section for clarity and completeness.
> Page 5 could use an example prompt from the dataset.
>
> Thank you for the suggestion. In the revised version, we added a concrete example of a prompt from our dataset on page 5, illustrating exactly how textual sections and figures are combined and presented to the model.
>
> Why exactly the authors chose the 4D printing as a scenario, if the authors mean to generalize the framework for any use-case, 2-3 related use-cases could be introduced.
> We selected 4D printing as a demonstration scenario because (i) it is a rapidly expanding and highly heterogeneous research field, (ii) knowledge is strongly fragmented across materials, stimuli, processes, and design strategies, and (iii) a baseline ontology (HERMES) already exists, enabling measurable enrichment and comparison. Although 4D printing is used as the main case study, the framework itself is domain-agnostic and can be applied to other areas such as polymer chemistry, biomedical materials, or advanced manufacturing.
>
> There is no evaluation of the generated ontology or the downstream task on the use of the ontology generated.
> We agree that downstream evaluation is important. In the revised manuscript, we added a new Section 4 that presents two ontology-driven use cases in smart materials and structures. These query–answer examples demonstrate how the enriched ontology supports material selection and composite architecture design, thereby showcasing its practical utility as a downstream decision-support tool.
>
> The paper is quite verbose at times with unnecessary details.
> Thank you for pointing this out. We shortened the introduction to LLMs, removed redundant low-level explanations of general LLM concepts, and consolidated several methodological descriptions to improve readability and focus on the core contributions.
>
> Many acronyms were repeatedly introduced since LLM was used for grammar correction possibly.
> In the revised version, we carefully reviewed the manuscript to ensure that each acronym is introduced only once and then reused consistently throughout the text.

---

### Meta-Review · Area_Chair_eU8M · 2025-12-23

**Summary:**

The reviewers consistently raised concerns about limited technical novelty, insufficient evaluation, and lack of strong baselines and ablations. While the proposed multimodal ontology enrichment framework is coherent and practically motivated, the paper was widely viewed as overly system-oriented, with weak empirical validation, unclear generalization beyond 4D printing, and missing reproducibility artifacts. These issues ultimately informed a negative recommendation.

**Reviewer Concerns:**

**Addressed by the authors:**

Improved clarity and presentation, including a shorter LLM introduction and clearer motivation.

Added example prompts, downstream query–answer use cases, and expert audit results.

Clarified the choice of 4D printing and argued for domain-agnostic design.

Provided partial evidence for the roles of multimodality, retrieval, fine-tuning, and validation.

**Still outstanding:**

No quantitative comparisons with competitive baselines.

No full ablation studies isolating each component.

Evaluation remains limited, relying on small expert audits and similarity-based metrics.

Domain generalization not empirically demonstrated.

Novelty remains questionable, with overlap to prior ontology enrichment pipelines.

Reproducibility concerns.

**Reviewer Scores:**

Reviewer J8DD: Likely unchanged (strong reject).

Reviewer oqR1: Possibly slightly improved, but still reject.

Reviewer yygp: Likely unchanged (reject).

Reviewer yZLL: Likely unchanged (strong reject).

---

### Decision · Program_Chairs · 2026-01-26

Reject